

# Metal complexation by histidine-rich peptides confers protective roles against cadmium stress in *Escherichia coli* as revealed by proteomics analysis

Patcharee Isarankura-Na-Ayudhya[1], Chadinee Thippakorn[2], Supitcha Pannengpetch[2], Sittiruk Roytrakul[3], Chartchalerm Isarankura-Na-Ayudhya[4], Nipawan Bunmee[1], Suchitra Sawangnual[1] and Virapong Prachayasittikul[4]

[1] Department of Medical Technology, Faculty of Allied Health Science, Thammasat University, Pathumthani, Thailand
[2] Center for Research and Innovation, Faculty of Medical Technology, Mahidol University, Bangkok, Thailand
[3] Genome Institute, National Center for Genetic Engineering and Biotechnology, National Science and Technology Development Agency, Pathumthani, Thailand
[4] Department of Clinical Microbiology and Applied Technology, Faculty of Medical Technology, Mahidol University, Bangkok, Thailand

Corresponding author
Chartchalerm Isarankura-Na-Ayudhya,
chartchalerm.isa@mahidol.ac.th

## ABSTRACT

The underlying mechanism and cellular responses of bacteria against toxic cadmium ions is still not fully understood. Herein, *Escherichia coli* TG1 expressing hexahistidine-green fluorescent protein (His6GFP) and cells expressing polyhistidine-fused to the outer membrane protein A (His-OmpA) were applied as models to investigate roles of cytoplasmic metal complexation and metal chelation at the surface membrane, respectively, upon exposure to cadmium stress. Two-dimensional gel electrophoresis (2-DE) and two-dimensional difference in gel electrophoresis (2D-DIGE) in conjunction with mass spectrometry-based protein identification had successfully revealed the low level expression of antioxidative enzymes and stress-responsive proteins such as manganese-superoxide dismutase (MnSOD; +1.65 fold), alkyl hydroperoxide reductase subunit C (AhpC; +1.03 fold) and DNA starvation/stationary phase protection protein (Dps; −1.02 fold) in cells expressing His6GFP in the presence of 0.2 mM cadmium ions. By contrarily, cadmium exposure led to the up-regulation of MnSOD of up to +7.20 and +3.08 fold in TG1-carrying pUC19 control plasmid and TG1 expressing native GFP, respectively, for defensive purposes against Cd-induced oxidative cell damage. Our findings strongly support the idea that complex formation between cadmium ions and His6GFP could prevent reactive oxygen species (ROS) caused by interaction between $Cd^{2+}$ and electron transport chain. This coincided with the evidence that cells expressing His6GFP could maintain their growth pattern in a similar fashion as that of the control cells even in the presence of harmful cadmium. Interestingly, overexpression of either OmpA or His-OmpA in *E. coli* cells has also been proven to confer protection against cadmium toxicity as comparable to that observed in cells expressing His6GFP. Blockage of metal uptake as a consequence of anchored polyhistidine residues on surface membrane limited certain amount of cadmium ions in which some portion could pass through and exert their toxic effects to cells as observed by the increased expression of

MnSOD of up to +9.91 and +3.31 fold in case of TG1 expressing only OmpA and His-OmpA, respectively. Plausible mechanisms of cellular responses and protein mapping in the presence of cadmium ions were discussed. Taken together, we propose that the intracellular complexation of cadmium ions by metal-binding regions provides more efficiency to cope with cadmium stress than the blockage of metal uptake at the surface membrane. Such findings provide insights into the molecular mechanism and cellular adaptation against cadmium toxicity in bacteria.

## INTRODUCTION

Cadmium ion is considered to be one of the harmful heavy metals that exerts its toxicities to all organisms including human, animals and microorganisms (*Vallee & Ulmer, 1972*). In microbes, the most common toxicity includes mutagenic effect, growth inhibition, physiological alterations and inhibition of enzymatic activities in metabolic pathway (*Bischoff, 1982*). Most of the microorganisms attempt to adapt themselves naturally from these metals by utilizing several cellular responses, e.g., reduction of metal-uptake process, sequestration of metal ions by biological macromolecules, transportation of metal ions out by efflux system, transformation of toxic metals by reductase enzymes, and metal sorption by biofilm and siderophores (*Giovanella et al., 2017*; *Nies, 1999*). Among these, active efflux has been thought to be the primary mechanism developed in prokaryotes to reduce intracellular cadmium ions. By contrast, intracellular complexation of the toxic cadmium ion by cadmium-binding components such as metallothioneins and phytochelatins is mainly used in eukaryotes (*Nies, 1992*).

In case of well-known bacteria e.g., *Escherichia coli* (*E. coli*), research findings during the past 50 years stated the toxic effects of cadmium ions and significant cellular responses and adaptation in various aspects as follows. (i) Cadmium toxicity: $Cd^{2+}$ is readily taken up by *E. coli* cells by means of an active transport system with a $K_m$ of approximately 2.1 μM (*Laddaga & Silver, 1985*). Exposure to cadmium ion results in long lag phase of cell proliferation, reduction of cell survival and loss of their ability to form colony (*Mitra, 1984*; *Mitra & Bernstein, 1978*; *Mitra et al., 1975*). Explanations for those phenomena could be accounted as the breakage of single-strand DNA, inhibition of metalloenzyme (i.e., alkaline phosphatase) and aggregation of cytoplasmic materials (*Mitra & Bernstein, 1978*; *Mitra et al., 1975*). Other toxic effects including disturbance of the cell division as a consequence of improper Z-ring formation, metabolic dysfunction, and imbalance of oxidative pressure leading to growth retardation were reported (*Hossain, Mallick & Mukherjee, 2012*). Induction of oxidative damage by superoxide radical together with inability to express protective enzymes and to replace damaged proteins by *de novo* protein synthesis have currently been reported to be the main reason for growth stasis and cell death in Cd poisoning (*Thomas & Benov, 2018*). (ii) Cellular responses and adaptation: the efflux

system not been widely accepted as the primary defense mechanism of *E. coli* to reduce the toxic effects of cadmium ions. Several reports stated that *E. coli* protected themselves against harmful cadmium mainly by synthesis of cytoplasmic cadmium-binding proteins to accumulate free $Cd^{2+}$ intracellularly (*Cohen, Bitan & Nitzan, 1991*; *Khazaeli & Mitra, 1981*; *Mitra, 1984*). Such an accumulation was proposed to control the internal environment and most likely to resume normal metabolic function after the long lag phase (*Khazaeli & Mitra, 1981*). Another group of proteins, namely cadmium-induced proteins (CDPs), was also produced as a stress-responsive protein during cadmium exposure (*Ferianc, Farewell & Nyström, 1998*). Among the CDPs, YodA protein (a protein containing histidine-rich N-terminal sequence; HGHHSH) has been extensively studied. It was found that the YodA protein mainly localized in both cytoplasm and periplasm but was only tranlocated into a periplasmic space during cadmium stress (*Puskárová et al., 2002*). It is believed that the role of YodA protein might be to decrease the intracellular concentration of cadmium ions by its ability to bind heavy metal (*Stojnev et al., 2007*). Moreover, the YodA protein is hypothesized to be part of a cadmium-exporting transport protein due to its sequence similarity to the C-terminal domain of a metal-binding receptor of a member of bacterial ATP-binding cassette transporters (*David et al., 2003*). In recent studies, the YodA has also been recognized as the ZinT, a metal-binding protein involved in zinc homeostasis (*Kershaw, Brown & Hobman, 2007*). Such an involvement of the histidine-rich peptide in the reduction of cadmium toxicity as well as in cadmium translocation strongly supports a significant role of histidine residue in metal regulation in bacterial cells. It seems that *E. coli* naturally utilizes both the intracellular complexation and metal transporting system to cope with cadmium stress but the proportion of which is not well understood. Currently, regulation of the CDPs together with the other general stress responses was subsequently studied either at the transcriptional or translational levels by gene-array technology or two-dimensional gel electrophoresis (*Brocklehurst & Morby, 2000*; *Ferianc, Farewell & Nyström, 1998*; *Isarankura-Na-Ayudhya et al., 2009*; *Lausova, Ferianc & Polek, 1999*). Results revealed that some regulatory systems, e.g., OxyR, SoxRS, RpoH, UspA, GrpE, RecA, and YodA, participated in the cadmium stress (*Ferianc, Farewell & Nyström, 1998*; *LaRossa, Smulski & Van Dyk, 1995*; *Lim et al., 2009*; *Puskárová et al., 2002*; *Shapiro & Keasling, 1996*; *Thomas & Benov, 2018*; *Van Dyk et al., 1995*). These inducible regulators can be classified in various groups as general stress regulons (UspA, YodA), SOS regulons (RecA), oxidative stress regulons (OxyR, SoxRS), heat shock protein regulons (RpoH) and chaperones (GrpE) (*Han & Lee, 2006*). It can be speculated that there will be some interconnection between these regulons in the cadmium stress.

In parallel to those long-term studies, genetic engineering has extensively been applied to construct metal-binding proteins in bacteria and other organisms. The most popular target belongs to a group of cysteine-rich peptides known as metallothionein (MT) that is ubiquitously found in various organisms. The expression of MTs had successfully conferred metal toleration and/or intracellular metal accumulation against cadmium stress in many cases (*He et al., 2014*; *Hou, Kim & Kim, 1988*; *Kille et al., 1990*; *Romeyer et al., 1988*). The thiol group of cysteine residues acts as soft Lewis base that preferably binds to cadmium ions (served as soft Lewis acid). However, the difficulty of controlling their reduced states limits

their maximum binding capability to cadmium ions (*Mahnam et al., 2018*). Moreover, the overexpression of proteins containing thiol-moiety is reported to alter the redox-regulating system and growth characteristics of *E. coli* (*Kondo et al., 2000*). Therefore, histidine residue (another amino acid known to coordinate with divalent ions) has widely been selected for cadmium binding and bioremediation purposes (*Nair & Robinson, 2001*; *Patel et al., 2010*; *Samuelson et al., 2000*; *Sousa, Cebolla & De Lorenzo, 1996*). The presence of histidine-rich peptides had conferred the engineered *E. coli* the capability to accumulate higher amounts of cadmium ions (∼11 fold) than that of the control (*Sousa, Cebolla & De Lorenzo, 1996*). Such findings explore the efficiency of applying metal-binding amino acids/peptides as tools not only to seek for their potential usages in bioremediation but also to study the molecular mechanism of metal toleration.

Herein, a proteomics profiling of *E. coli* TG1 upon exposure to sub-lethal dose of cadmium ions has been analyzed by using Two-dimensional gel electrophoresis (2-DE) and two-dimensional difference in gel electrophoresis (2D-DIGE) in conjunction with mass spectrometry-based protein identification. Engineered *E. coli* cells expressing metal-binding regions on the outer membrane or cytoplasm have been utilized in order to test whether controlling of metal transport across the membrane or intracellular metal complexation played major roles in modulating cellular adaptations against toxic cadmium ions. In Particular, the expression of green fluorescent protein carrying hexahistidine (His6GFP) or polyhistidine fused to outer membrane protein A (His-OmpA) have been used as models to mimic metal-binding regions located in the cytoplasm and on the surface membrane, respectively. It has previously been reported that the expression of chimeric His6GFP could modulate metal homeostasis and mobility inside *E. coli* cells (*Isarankura-Na-Ayudhya et al., 2005*). The rationale behind the expression of histidine-rich peptide in the cytoplasm is expected to chelate cadmium ions in order to prevent the sequelae of cadmium-induced oxidative cell damage *via* reactive oxygen species. The expression of His-OmpA has also been found to provide high binding affinity for metal ions at the surface of *E. coli* (*Isarankura-Na-Ayudhya et al., 2005*). The presence of polyhistidine on the cell surface is expected to trap cadmium ions and limit certain amount of cadmium to transport across the membrane. Then, the investigation of differentially-expressed proteins between these engineered *E. coli* and control cells has been conducted in order to gain a better understanding of the alternative ways that bacteria acclimatize themselves in toxic environments as compared to those observed in nature.

# MATERIALS AND METHODS

## Bacterial strain and plasmids

*E. coli* strain TG1 (*sup*E, *hsd*Δ5, *thi*Δ(lac-proAB), F'[*tra*D36 *pro*AB+ *lac*I�q lacZΔM15]) was used as host cell. To express cytoplasmic metal-binding motifs, a plasmid designated as pHis6GFPuv (*Isarankura-Na-Ayudhya et al., 2005*) that encodes a chimeric green fluorescent protein carrying hexahistidine was transformed and further expressed in host cells. Cells harboring pUC19 and pGFPuv (Clontech Laboratories, Mountain View, CA, USA) were used as controls. In parallel, a plasmid namely pEVZn (*Mejare, Ljung &*

*Bulow, 1998*) that codes for the outer membrane protein A-polyhistidine fusion protein was expressed in order to generate cells expressing surface metal-binding motifs. For comparison, cells harboring pEV208 (*Mejare, Ljung & Bulow, 1998*) was used to over-express the outer membrane protein.

## Growth patterns of cells in the presence of sub-lethal dose of cadmium ions

All of the aforementioned cells with the exception of *E. coli* host were grown in 5 ml Luria-Bertani (LB) broth (10 g/L tryptone, 5 g/L NaCl and 5 g/L yeast extract, pH 7.2) that is supplemented with 100 mg/L ampicillin at 37 °C, 150 rpm for overnight. The cultivation of the TG1 host was performed in a similar manner but without the addition of ampicillin. Then, 50 µl of overnight cultures were transferred into 5 ml broth and grown until $OD_{600}$ reached 0.5. Cells were adjusted to equal optical density at 0.05 in 5 ml LB broth supplemented with 100 mg/L ampicillin. Dose response assay was performed by addition of cadmium chloride stock solution to yield final concentrations of 0–1.0 mM. Cultures were further incubated at 37 °C, 150 rpm for 12 h. The growth rate was determined by monitoring the absorbance at 600 nm by spectrophotometer. It is noteworthy that the sub-lethal dose of 0.2 mM cadmium ions (exhibiting ∼50% growth inhibition of TG1 host) was selected as an effective dose for further experiments.

## Two-dimensional gel electrophoresis (2-DE) of crude protein extracts

Overnight cultures of cells in 5 ml LB or LB/Amp were adjusted to $OD_{600}$ of 1.0. One milliliter of cell suspension was inoculated into 50 ml LB or LB/Amp. Cells were grown at 37 °C, 150 rpm for 3 h prior to addition of $CdCl_2$ to yield the final concentration of 0.2 mM. Cultivation was continued at 37 °C, 150 rpm for 15 h. Cells were collected by centrifugation at 4 °C, 6,000 rpm for 10 min. Crude protein extracts were prepared as previously described (*Isarankura-Na-Ayudhya et al., 2010*). The protein concentration was measured by Bradford's method (Bio-Rad protein assay; Bio-Rad Laboratories, Hercules, CA, USA). These protein extracts were subjected to two-dimensional gel electrophoresis as follows. Three hundred micrograms of protein extract was mixed thoroughly with 250 µl of rehydration buffer (8 M urea, 4% CHAPS, 2 mM TBP, 0.001% bromphenol blue, 2.8 mg/ml dithiothreitol and 12 µl/ml destreak) containing 1% 3–10 IPG buffer and stored at room temperature for 10 min. Removal of insoluble material was further performed by spinning at 13,000 rpm for 10 min at 20 °C. The 13-cm IPG strips (pH range of 3–10) were placed in the IPGphor strip holder gel-side-down in rehydration solution containing sample proteins. These strips were then covered with mineral oil. Samples were run through steps of strip rehydration (30 V, 12 h) and isoelectric focusing (500 V for 1 h, 1,000 V for 1 h, and 8,000 V to reach 16,000 V h). The maximum current was maintained at 50 mA per strip. Once complete, the strip was equilibrated for two times (15 min each) in equilibration buffer (50 mM Tris pH 8.8, 6 M urea, 30% glycerol, 2% SDS, 0.03% bromphenol blue) supplemented with 65 mM DTT and 135 mM iodoacetamide. The separation of protein in the second dimension was performed using Hoefer[TM] DALT on 12.5% SDS-polyacrylamide gels. Proteins were separated under applied voltage of 250

V, 10 mA per strip at 20 °C for 30 min following by 250 V, 20 mA per strip at 20 °C for 2.5 h until the bromphenol blue dye front reached 0.5 cm from the bottom of the gel. Gels were further stained with colloidal Coomassie brilliant blue G for overnight. Excess dye was removed by rinsing several times with deionized distilled water. Gels were scanned with the Canoscan LiDE20 scanner (Canon, Huntington, NY, USA). Intensity of protein spots was analyzed by the ImageJ software tool (*Schneider, Rasband & Eliceiri, 2012*).

## Protein labeling and Two-Dimensional Difference in Gel Electrophoresis (2D-DIGE)
### Sample preparation
Protein extracts were cleaned-up by using the 2-D Clean-up kit (GE Healthcare, Chicago, IL, USA). Then, protein pellets were dissolved in a buffer containing 30 mM Tris–HCl, 7 M urea, 2 M thiourea, 4% (w/v) CHAPS, pH 8.5. Proteins were subsequently labeled with CyDye using GE Cydye DIGE Fluor (minimal dyes) Labeling Kit (GE Healthcare, Chicago, IL, USA) according to the manufacturer's instruction. Then, 50 μg of protein from each sample was pooled together as the internal standard. In parallel, a total of 50 μg protein of each sample was randomly labeled with 400 pmol Cy3 or Cy5, and the internal standard was labeled with Cy2 for 30 min on ice. Samples were then rehydrated into 18 cm IPG strips (pH 3–10 NL) (GE Healthcare, Chicago, IL, USA) overnight in rehydration buffer (8 M urea, 4% CHAPS, 13 mM dithiothreitol, and 1% IPG buffer 3–10 NL). Isoelectric focusing was carried out using an IPGphor III apparatus (GE Healthcare, Chicago, IL, USA) according to the following procedures: 500 V for 500 V h, 1,000 V for 800 V h, and 10,000 V to reach 36,000 V h. Strips were further equilibrated for 15 min in equilibration buffer (75 mM Tris–HCl pH 8.8, 6 M urea, 30% glycerol, 2% SDS, 0.002% bromphenol blue) containing 1% DTT and then for 15 min with 2.5% iodoacetamide. Equilibrated IPG strips were transferred onto 12.5% SDS-polyacrylamide gels that had been pre-casted in low fluorescence glass plates. The second dimension separation was then carried out at 10 W/gel using Ettan Dalt six electrophoresis system (GE Healthcare, Chicago, IL , USA).

### Gel image analysis
The 2D-DIGE gels were visualized using a Typhoon TRIO fluorescence scanner (GE Healthcare, Chicago, IL, USA) with excitation/emission at 532/580 nm (Cy3), 633/670 nm (Cy5) and 488/520 nm (Cy2). Scanning resolution used was 100 μm with photomultiplier of 550 V. Gel image and statistical analyses were performed using the DeCyder™ 2D Differential Analysis Software (DeCyder 2D version 7.2; DeCyder, Amersham, UK) by differential in-gel analysis (DIA) and biological variation analysis (BVA). Gels were subjected to silver staining for spot visualization and picking.

## Peptide mass fingerprinting (PMF) analysis
Peptide mass fingerprinting (PMF) analysis of the protein separated by 2-DE was performed as previously described (*Isarankura-Na-Ayudhya et al., 2010*). The experiment was initiated by cutting spots of protein from gels and these spots were further transferred to 96-well microtitre plate and then soaked in 50% methanol and 5% acetic acid for overnight. In-gel digestion of protein was performed by the addition of sequencing grade of modified

trypsin (Promega, Southampton, UK). Digested peptide fragments were extracted from gel segments on a Spot Handling Workstation (GE Healthcare, Chicago, IL, USA) using preset protocols from the manufacturer. Protein identification was further carried out using MALDI-TOF mass spectrometer (Model ReflexIV; Bruker Daltonics, Billerica, MA, USA) based on peptide fingerprint map. The tryptic digested peptide was mixed with a solution of 10 mg/ml $\alpha$-cyano-4-hydroxycinnamic acid (LaserBio Labs, Sophia-Antipolis Cedex, France) in 66% acetonitrile and 0.1% trifluoroacetic acid (TFA) and further spotted onto a 96-well target plate. The acquisition of mass spectra was conducted in the positive ion reflector delayed extraction mode using approximately 200 laser shots. Creation of peak lists was performed using the XMASS software (Bruker Daltonics, FRG). These peaks were then queried using the MASCOT search engine (MatrixScience, http://www.matrixscience.com/) for protein identification. The reference database used in this study was NCBInr 20130802 (31350673 sequences; 10834990394 residues). The searching criteria were as follows: complete carbamidomethylation of cysteine and partial methionine oxidation; an initial mass tolerance of $\pm 1.2$ Da; the number of missed cleavage sites of up to 1. The accuracy of the experimental to theoretical $p$I as well as the molecular weight of proteins was carefully taken into consideration.

## Protein network analysis

Identified proteins were queried using the STRING software version 9.05 (http://string-db.org/) in order to create functional protein association networks. This database weights and integrates direct (physical) and indirect (functional) associations from various sources, e.g., genetic context, high-throughput experiments, co-expression and previous knowledge.

## Statistical analysis

For the 2D-DIGE gel analysis, only the spots with changes having an abundance ratio of 1.5 fold and $P$ values $< 0.05$ (Student's $t$-test) were marked and selected for further protein identification. For the PMF analysis, a search result score greater than 71 was considered to be of significant difference ($p < 0.05$).

# RESULTS

## Expression of green fluorescent protein of *E. coli* expressing cytoplasmic histidine-rich protein (chimeric His6GFP)

*E. coli* TG1 was applied as a host for the transformation of pUC19 (control plasmid), pGFPuv and pHis6GFPuv. Expression of the green fluorescent protein was confirmed by observing the bright greenish fluorescence of colonies of *E. coli* expressing native GFP or His6GFP under illumination with UV light (Figs. 1C and 1D). Meanwhile, no fluorescence was detected for TG1 host and TG1 carrying pUC19 (Figs. 1A and 1B).

## Effect of cadmium stress on *E. coli* expressing cytoplasmic histidine-rich protein (chimeric His6GFP)

In the absence of cadmium ions, it seems that the TG1 carrying a control plasmid (Fig. 2A) growed up in the LB broth faster than the others. A long lag phase ($\sim 3.5$–4 h) could be detected in the cells expressing native GFP (Fig. 2B). The growth pattern of cells expressing
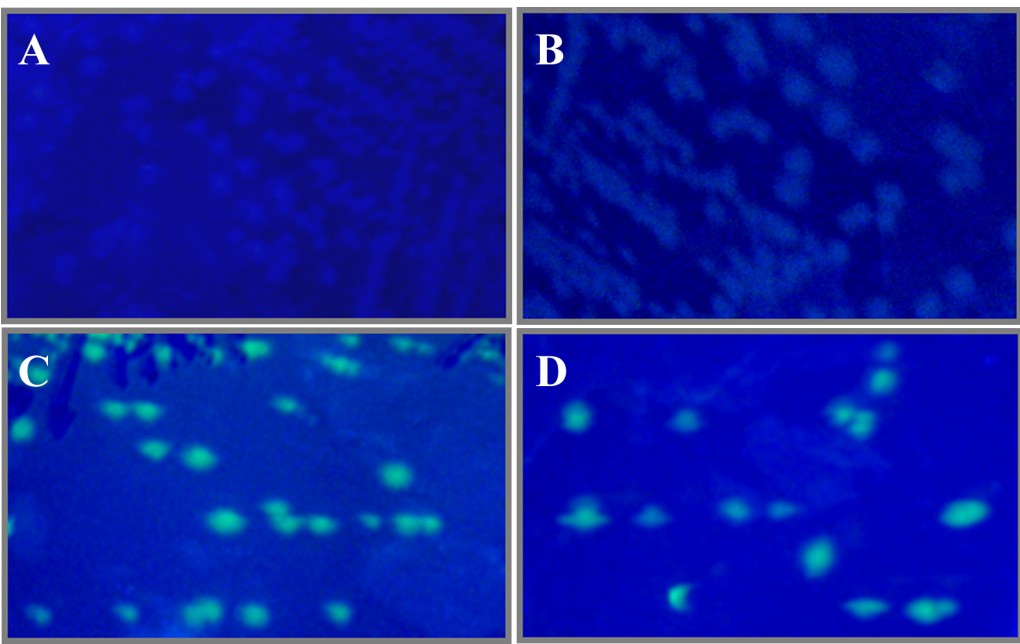

**Figure 1** **Expression of GFP in tested cells as deduced from UV illumination.** Non-fluorescent colonies of TG1 host (A) and TG1 carrying pUC19 (B) as well as greenish fluorescent colonies of *E. coli* expressing native GFP (C) or His6GFP (D).

His6GFP (Fig. 2C) resembled those in between the TG1 carrying pUC19 (Fig. 2A) and the native GFP-expressing cells (Fig. 2B).

In the presence of effective dose of cadmium, a growth arrest could be observed in cells expressing native GFP (Fig. 2B). However, this suppressing effect was more pronounced in cells carrying pUC19 (Fig. 2A). Interestingly, cells expressing the hexahistidine-GFP fusion protein displayed similar growth rates in the presence and absence of cadmium ions (Fig. 2C).

## Proteomics profiling of *E. coli* expressing cytoplasmic histidine-rich protein (chimeric His6GFP)

Experimentation was initiated *via* the preparation of a reference map of the *E. coli* host proteome (Fig. 3). Approximately 200–250 spots of protein could be detected using a 13-cm IPG strip with pH range of 3–10. Some of these spots were picked up for further identification *via* peptide mass fingerprinting (http://www.matrixscience.com) as shown in Table 1. Most of the identified proteins pertained to energy metabolism, chaperones, heat shock proteins or stress proteins, transporters, protein synthesis machinery, and outer membrane proteins.

Further experiments were conducted to address the functional roles of cytoplasmic histidine-rich protein (His6GFP) on cellular adaptation against toxic cadmium ions. Therefore, protein expression profiles of *E. coli* expressing His6GFP grown in the absence or presence of 0.2 mM $CdCl_2$ (Fig. 4C) were compared with cells harboring pUC19 and cells expressing native GFP (as represented in Figs. 4A–4B, respectively). The effects

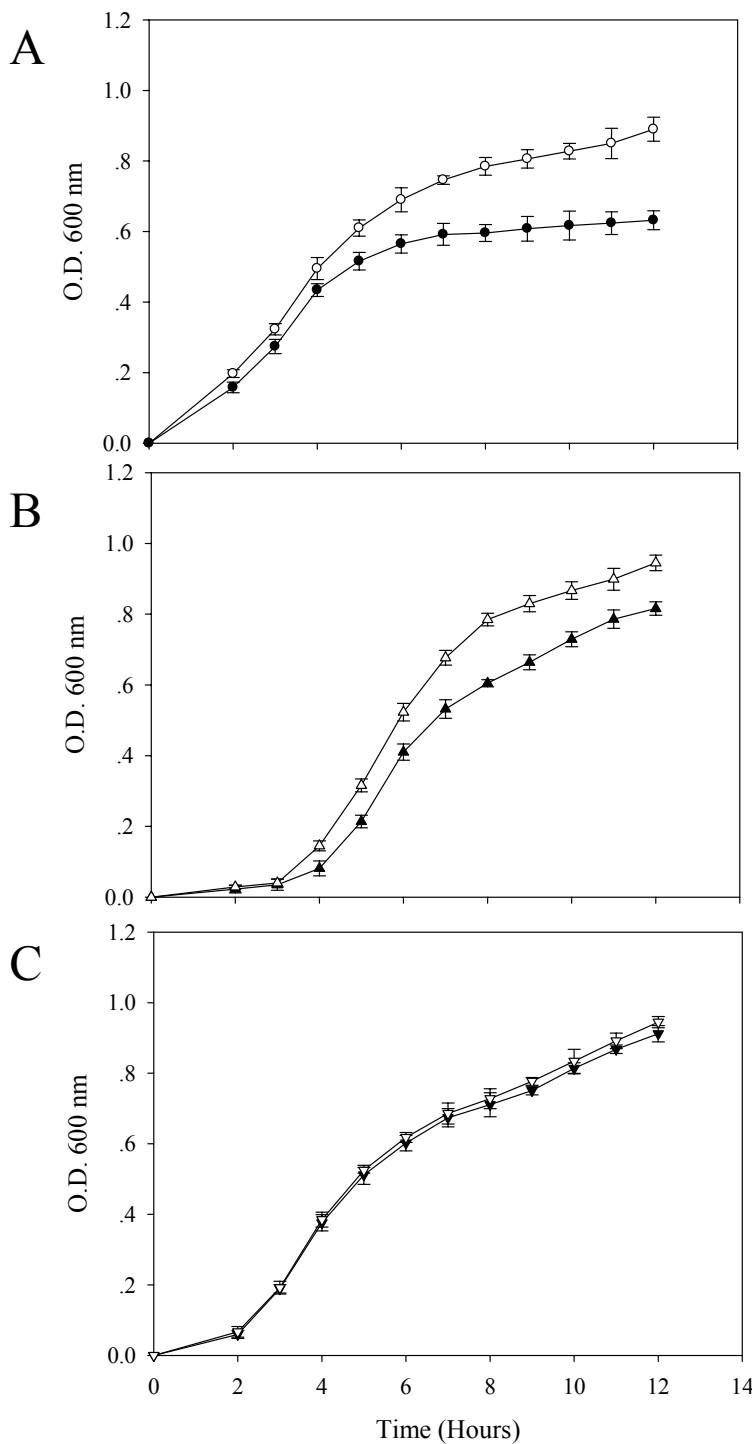

**Figure 2 Effect of cadmium stress on *E. coli* expressing chimeric His6GFPuv, cells expressing native GFP and cells carrying control plasmid (pUC19).** Growth patterns of TG1/pUC19 (A), TG1/pGFPuv (B) and TG1/pHis6GFPuv (C) in the absence (opened symbol) or presence (closed symbol) of 0.2 mM cadmium ions.

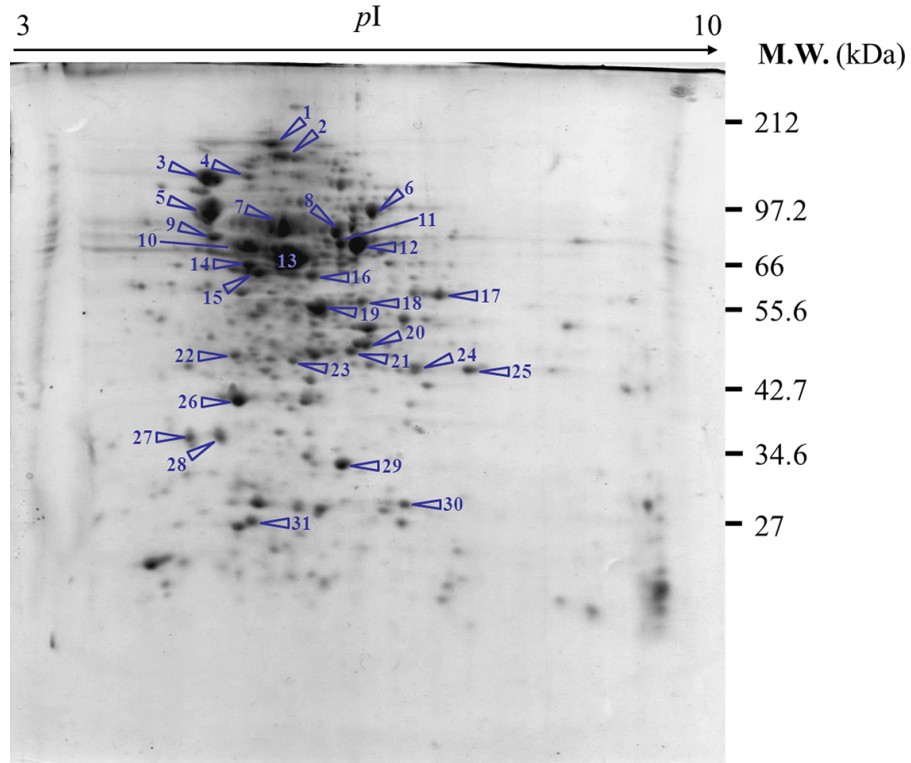

**Figure 3** **Reference map representing protein profiles of *E. coli* TG1.** Numbers of protein spot represents identified proteins as presented in Table 1.

of cadmium ions on the expression of various proteins were scrutinized as follows. Aconitase, an enzyme implicated in the Tricarboxylic acid (TCA) cycle, was down-regulated and disappeared in cells expressing His6GFP. Translation elongation factor-Ts was up-regulated in all cases (at 2.12, 1.84 and 1.35 fold for TG1/pUC19, TG1/pGFPuv and TG1/pHis6GFPuv, respectively) (Table 2). Glycerol kinase, an enzyme involved in glycerol uptake and lipolysis, was up-regulated in cells expressing His6GFP (+1.92 fold) and native GFP (+1.73 fold). Oligopeptide-binding protein was down-regulated at −1.65- and −1.34 fold for control cells and cells expressing His6GFP while up-regulation was found in the case of GFP (+1.37 fold). The H⁺-transporting ATPase was up-regulated at 2.69 and 1.30 fold in case of control cells and cells expressing His6GFP, respectively. More importantly, the presence of cytoplasmic histidine residue due to the expression of His6GFP was found to play imperative roles in stress defense mechanisms. Manganese-superoxide dismutase (MnSOD), an oxidative scavenging enzyme, was found to be down-regulated in cells expressing His6GFP (−1.65 fold). Alkyl hydroperoxide reductase, a thiol-specific antioxidant protein, was up-regulated in the case of native GFP-expressing cells (+1.71 fold). This is in contrast to those observed in the case of His6GFP (−1.10 fold). For membrane proteins, our results revealed the down-regulation of OmpC precursor (−1.15, −1.31 and −1.56 fold) and OmpF (−1.18 fold and absence) in all cases in response to cadmium ions.

**Table 1 Proteins of *Escherichia coli* TG1 host identified by mass spectrometry and peptide mass fingerprinting (PMF) analysis.**

| Spot no. | Accession no. | Description | Calculated $pI$ value | Norminal mass ($M_r$) | Protein score | Sequence coverage (%) |
|---|---|---|---|---|---|---|
| 1 | gi\|485870954 | Aconitate hydratase 2 | 5.22 | 91,065 | 80 | 18 |
| 2 | gi\|486356887 | Chaperone ClpB | 5.37 | 95,712 | 76 | 31 |
| 3 | gi\|446438287 | Molecular chaperone DnaK | 4.83 | 69,096 | 115 | 46 |
| 4 | gi\|518680918 | Heat shock protein 90 | 5.09 | 71,360 | 84 | 27 |
| 5 | gi\|446651775 | Molecular chaperone GroEL | 4.85 | 57,494 | 89 | 47 |
| 6 | gi\|486650025 | Periplasmic oligopeptide-binding protein | 6.06 | 62,730 | 114 | 49 |
| 7 | gi\|485671395 | Glycerol kinase | 5.97 | 59,776 | 109 | 48 |
| 8 | gi\|1311039 | Chain A, Dipeptide binding protein | 5.75 | 57,599 | 79 | 35 |
| 9 | gi\|485778118 | ATP synthase F1, beta subunit | 4.90 | 50,341 | 125 | 41 |
| 10 | gi\|33383669 | Isocitrate dehydrogenase | 5.33 | 43,192 | 84 | 35 |
| 11 | gi\|485734442 | Tryptophanase | 5.80 | 40,109 | 81 | 54 |
| 12 | gi\|485734442 | Tryptophanase | 5.80 | 40,109 | 108 | 72 |
| 13 | gi\|445923422 | Elongation factor-Tu | 5.07 | 41,052 | 89 | 44 |
| 14 | gi\|485752584 | Phosphoglycerate kinase | 5.02 | 40,719 | 104 | 59 |
| 15 | gi\|485747310 | Glycerophosphodiester phosphodiesterase | 5.15 | 36,482 | 79 | 51 |
| 16 | gi\|485696065 | Fructose-bisphosphate aldolase | 5.52 | 39,337 | 88 | 41 |
| 17 | gi\|485665932 | Glyceraldehyde-3-phosphate dehydrogenase | 6.33 | 36,204 | 71 | 43 |
| 18 | gi\|9507742 | Outer membrane protein P | 5.91 | 35,477 | 88 | 42 |
| 19 | gi\|485723954 | Malate dehydrogenase | 5.62 | 32,532 | 72 | 48 |
| 20 | gi\|510898944 | D-ribose-binding periplasmic protein | 6.85 | 30,949 | 72 | 50 |
| 21 | gi\|486435677 | Uridine phosphorylase | 5.71 | 27,262 | 98 | 68 |
| 22 | gi\|485953066 | Translation elongation factor-Ts | 5.18 | 28,505 | 74 | 39 |
| 23 | gi\|487374764 | Purine nucleoside phosphorylase | 5.42 | 26,157 | 78 | 34 |
| 24 | gi\|485673269 | Glyceraldehyde-3-phosphate dehydrogenase | 6.32 | 31,252 | 74 | 36 |
| 25 | gi\|447012380 | Amino acid ABC transporter substrate-binding protein | 7.74 | 27,159 | 73 | 42 |
| 26 | gi\|170172436 | Alkyl hydroperoxide reductase C22 protein | 4.92 | 19,250 | 92 | 56 |
| 27 | gi\|487464714 | Glucose-specific phosphotransferase enzyme | 4.73 | 18,198 | 87 | 43 |
| 28 | gi\|446006533 | Lipid hydroperoxide peroxidase | 4.75 | 17,952 | 76 | 61 |
| 29 | gi\|485651470 | DNA starvation/stationary phase protection protein Dps | 5.72 | 18,711 | 85 | 69 |
| 30 | gi\|486164894 | 50S ribosomal protein L9 | 6.17 | 15,772 | 78 | 48 |
| 31 | gi\|446245719 | Universal stress protein A | 5.11 | 16,086 | 71 | 70 |

## Functional roles of histidine residue fused with outer membrane protein in *E. coli*

It has previously been proven that the presence of polyhistidine on the surface membrane of such engineered *E. coli* provided high binding affinity to zinc ions of up to 60 fold (*Isarankura-Na-Ayudhya et al., 2005*). Therefore, these cells were applied in the current study in order to reduce the cell permeability of cadmium ions. Figure 5 demonstrates the growth patterns of TG1/pEV208 and TG1/pEVZn in the presence and absence of cadmium ions. Our results revealed that the overexpression of the outer membrane protein of cells carrying plasmid pEV208 could protect the cell growth from the hazardous effect of cadmium ions. More importantly, the expression of polyhistidine-OmpA of cells

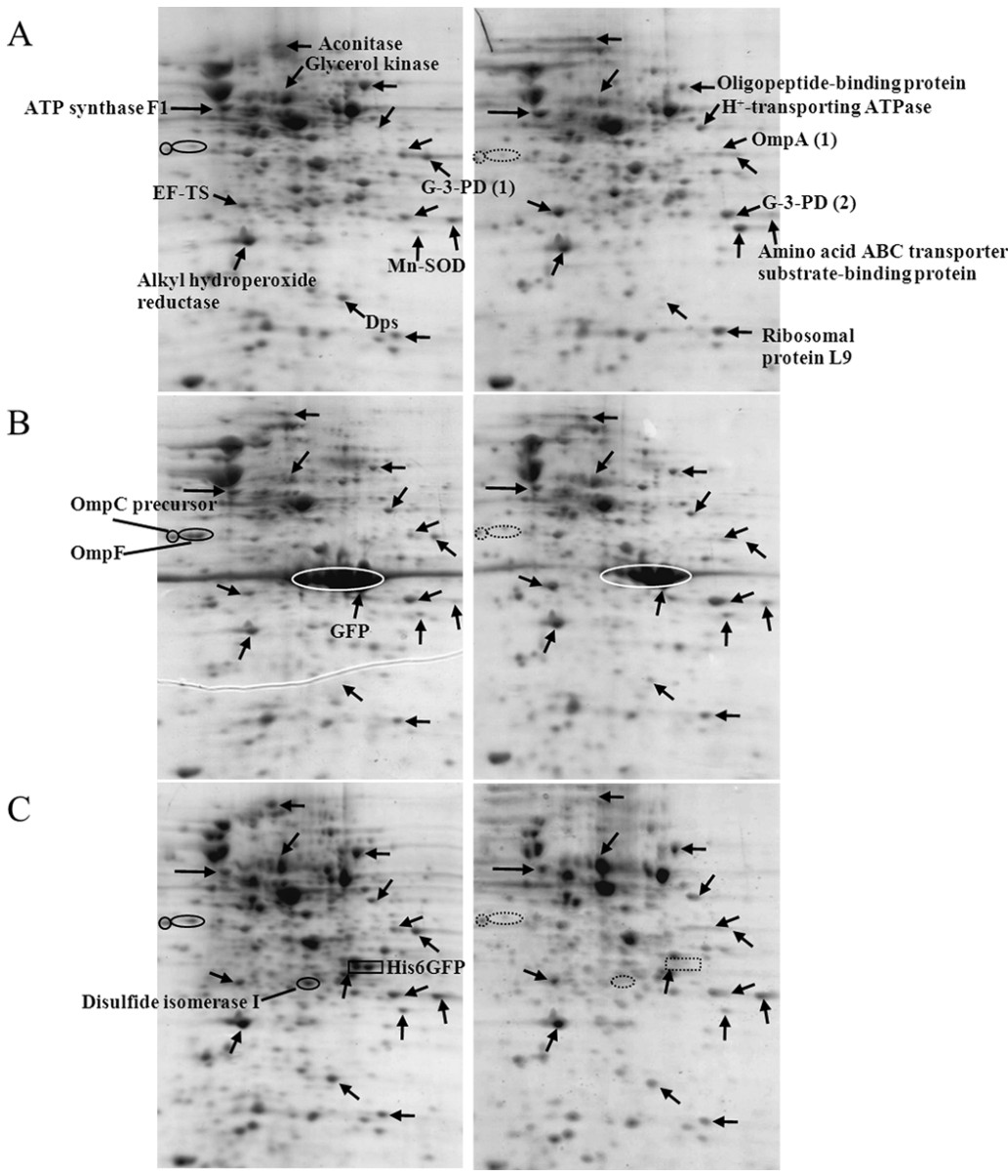

**Figure 4 Protein expression profiles of *E. coli* grown in the presence of 0.2 mM cadmium ions.**
TG1/pUC19 (A), TG1/pGFPuv (B) and TG1/pHis6GFPuv (C).

harboring pEVZn could mediate a little bit higher degree of cell division than that of cells expressing the OmpA alone. Results from the proteomics analysis revealed that the enzyme glyceraldehyde-3-phosphate dehydrogenase form 1 was up-regulated in cells expressing OmpA (+2.69 fold) while the disappearance of this enzyme was observed in the case of cells expressing His-OmpA (Fig. 6 and Table 2). A diverse phenomenon was detected on the regulation of glyceraldehyde-3-phosphate dehydrogenase form 2 at −2.29 and +1.63 fold for TG1/OmpA and TG1/His-OmpA, respectively. Furthermore, H⁺-transporting ATPase was down-regulated in the case of the control (−2.64 fold) while no change was found in

**Table 2  Changes of differentially expressed proteins of *E. coli* TG1 following exposure to cadmium.**

| Protein | Up(+)/Down(−) regulation (fold) | | | | |
|---|---|---|---|---|---|
| | TG1 carrying pUC19 plasmid | TG1 expressing GFP | TG1 expressing His6GFP | TG1 expressing OmpA | TG1 expressing His-OmpA |
| **Energy metabolism** | | | | | |
| *Glycolysis* | | | | | |
| Glyceraldehyde-3-phosphate dehydrogenase (1) | −2.02 | −1.84 | Absence | +2.69 | Absence |
| Glyceraldehyde-3-phosphate dehydrogenase (2) | +1.75 | +1.65 | −1.06 | −2.29 | +1.63 |
| *Tricarboxylic acid cycle* | | | | | |
| Aconitase | −1.20 | No change | Absence | No change | Absence |
| *Glycerol uptake and lipolysis* | | | | | |
| Glycerol kinase | −1.60 | +1.73 | +1.92 | +1.43 | +1.33 |
| **Protein biosynthesis machinery** | | | | | |
| Translation elongation factor-Ts | +2.12 | +1.84 | +1.35 | −0.39 | +1.37 |
| **Transporters** | | | | | |
| Oligopeptide-binding protein | −1.65 | +1.37 | −1.34 | +1.22 | +1.21 |
| Amino acid ABC transporter substrate-binding protein | −1.42 | +1.37 | +1.29 | Not detectable | +1.24 |
| H+-transporting ATPase | +2.69 | −1.09 | +1.30 | −2.64 | No change |
| ATP synthase F1, beta subunit | +1.32 | +1.26 | +1.08 | +1.86 | −1.44 |
| **Stress defense mechanism** | | | | | |
| MnSOD | +2.20 | +1.29 | −1.65 | Not detectable | +2.08 |
| Alkyl hydroperoxide reductase | −1.15 | +1.71 | −1.10 | +1.52 | −1.01 |
| DNA starvation/stationary phase protection protein; Dps | −3.29 | Low amount | −1.35 | Low amount | −1.05 |
| **Others** | | | | | |
| Ribosomal protein L9 | +2.08 | +1.33 | −1.29 | +1.42 | +1.57 |
| GFP | Not detectable | −1.45 | Not detectable | Not detectable | Not detectable |
| His6GFP | Not detectable | Not detectable | −2.24 | Not detectable | Not detectable |
| Disulfide isomerase I | −1.71 | Not determined | −1.85 | Not determined | −1.36 |
| OmpA (1) | −1.68 | −1.26 | −1.40 | −1.35 | Absence |
| OmpC precursor | −1.15 | −1.31 | −1.56 | +1.32 | +2.11 |
| OmpF | −1.18 | Absence | Absence | No change | Absence |

cells expressing His-OmpA. In addition, cadmium ions stimulated expression of MnSOD in cells expressing His-OmpA (+2.08 fold). It is noteworthy that an increased amount of the OmpC precursor was observed in cells expressing His-OmpA (+2.11 fold) as well as in cells expressing OmpA alone (+1.32 fold). Such a finding is in contrast to those observed in the case of cells expressing His6GFP and others as abovementioned.

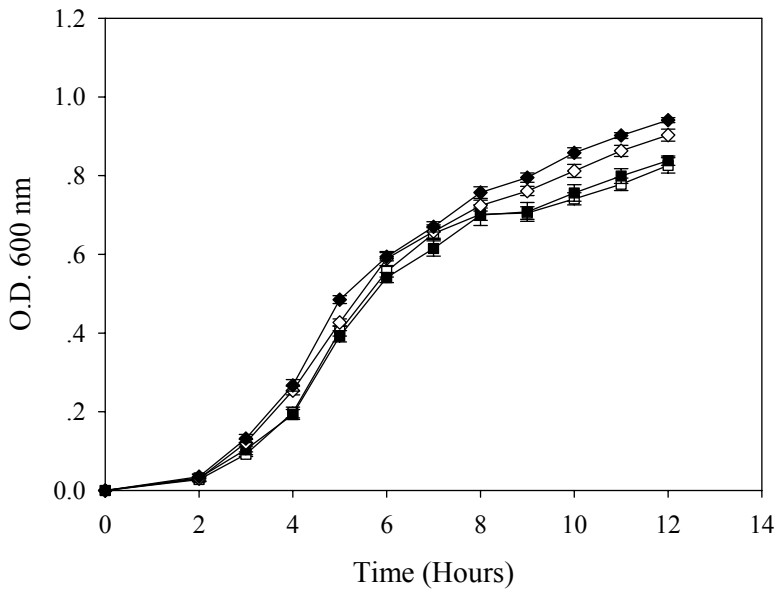

**Figure 5** **Growth patterns of *E. coli* grown in the presence of 0.2 mM cadmium ions.** TG1/pEV208 (square) and TG1/pEVZn (diamond) in the absence (opened symbol) or presence (closed symbol) of 0.2 mM cadmium ions.

## Comparison of cadmium-induced protein alterations in engineered *E. coli* expressing cytoplasmic polyhistidine and polyhistidine-anchored surface membrane

Differentially expressed proteins of engineered *E. coli* upon exposure to cadmium ions were quantitatively identified as summarized in Table 2. For enzymes implicated in energy metabolism, glyceraldehyde-3-phosphate dehydrogenase form 1 was not detected in all cases except in cells expressing OmpA. The up-regulation of glyceraldehyde-3-phosphate dehydrogenase form 2 was found in TG1 carrying pUC19, cells expressing native GFP and cells expressing His-OmpA at 1.75, 1.65 and 1.63 fold, respectively. Aconitase was down-regulated in most cases. Meanwhile, the up-regulation of glycerol kinase was observed in the range of 1.3–1.9 fold in all cases except for the control. For the protein synthesis machinery, all cell types (with the exception of TG1 expressing OmpA) displayed approximately 1.3–2.1 fold increases. For transporters, oligopeptide-binding protein was decreased ~1.3–1.6 fold only in the case of cells expressing His6GFP and control cells. Increased amounts of amino acid ABC transporter substrate-binding protein of 1.2–1.3 fold were found in TG1 expressing native GFP, His6GFP and His-OmpA while no change was observed in control cells. In regard to the $H^+$-transporting ATPase, overexpressing of up to 2.7 fold was found in control cells. Up-regulation of approximately 1.9 fold of ATP synthase F1 (membrane-bound ATP synthase) was notified in cells expressing OmpA. For the stress defense mechanism, MnSOD is detected to be the most important antioxidative enzyme responsible for the toxic cadmium. Most cells enhanced the production of MnSOD in the range of 1.29–2.20 fold. Importantly, down-regulation of MnSOD (~1.65 fold) was found only in cells expressing cytoplasmic polyhistidine. It should be noted that the

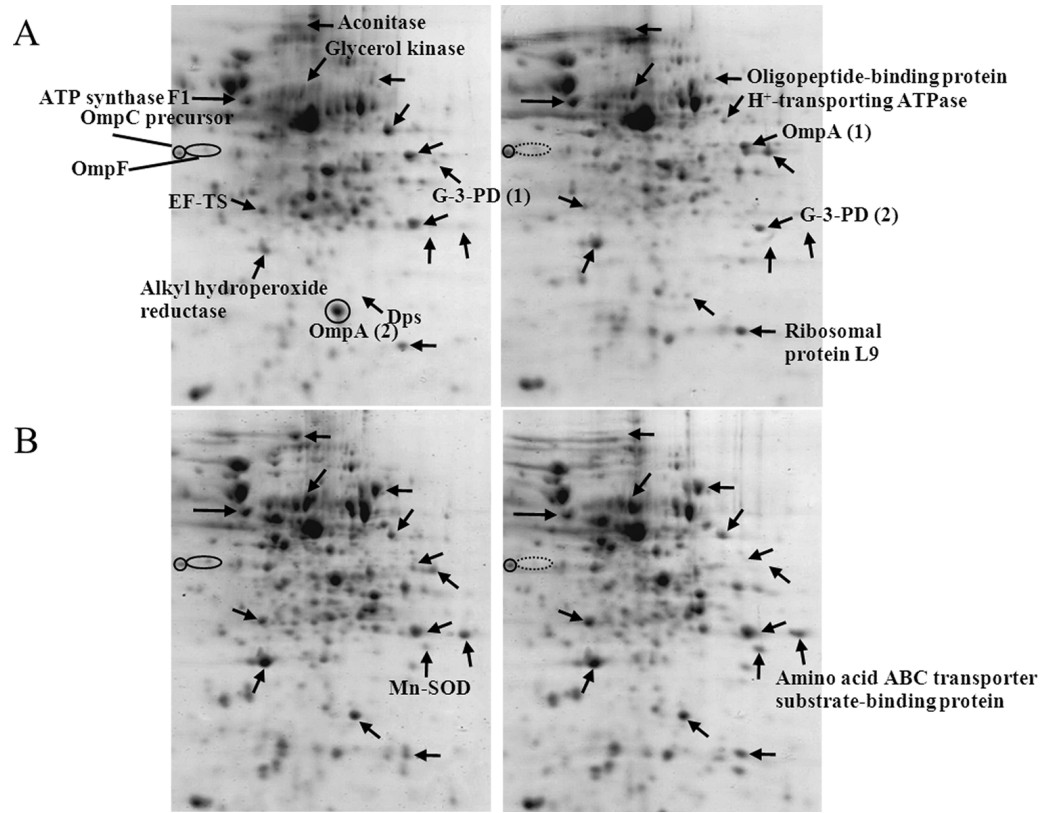

**Figure 6  Protein expression profiles of *E. coli* grown in the presence of 0.2 mM cadmium ions.** TG1/pEV208 (A) and TG1/pEVZn (B).

up-regulation of AphC of up to 1.71 and 1.52 was found in TG1 expressing native GFP and cells expressing only outer membrane protein. For the ribosomal protein L9, up-regulation in the range of 1.3–2.1 fold was found in all cases with the exception of TG1 expressing His6GFP. Moreover, an increased expression of the OmpC precursor of approximately 1.3–2.1 fold as detected in cells expressing OmpA and His-OmpA. The disappearance of OmpF was observed in TG1 expressing native GFP, His6GFP and His-OmpA.

## Confirmation of cadmium-induced protein alterations using two-dimensional difference in gel electrophoresis (2D-DIGE)

To further confirm the protective roles of metal complexation by histidine-rich peptides against cadmium stress, quantitative proteomics investigation using 2D-DIGE was employed as to provide high sensitivity and high discrimination power on protein expression. As shown in Fig. 7A, three different samples were individually labeled with three fluorescent dyes (Cy2, Cy3 and Cy5) and co-separated in one gel, which mediates spot matching and quantitation in a simpler and more accurate manner. An internal standard, as derived from a mixture of all samples pooled together and labeled with Cy2, was created to facilitate the normalization of each spot among all gels. This provides a reliable and reproducible detection by minimizing gel-to-gel variation and increasing

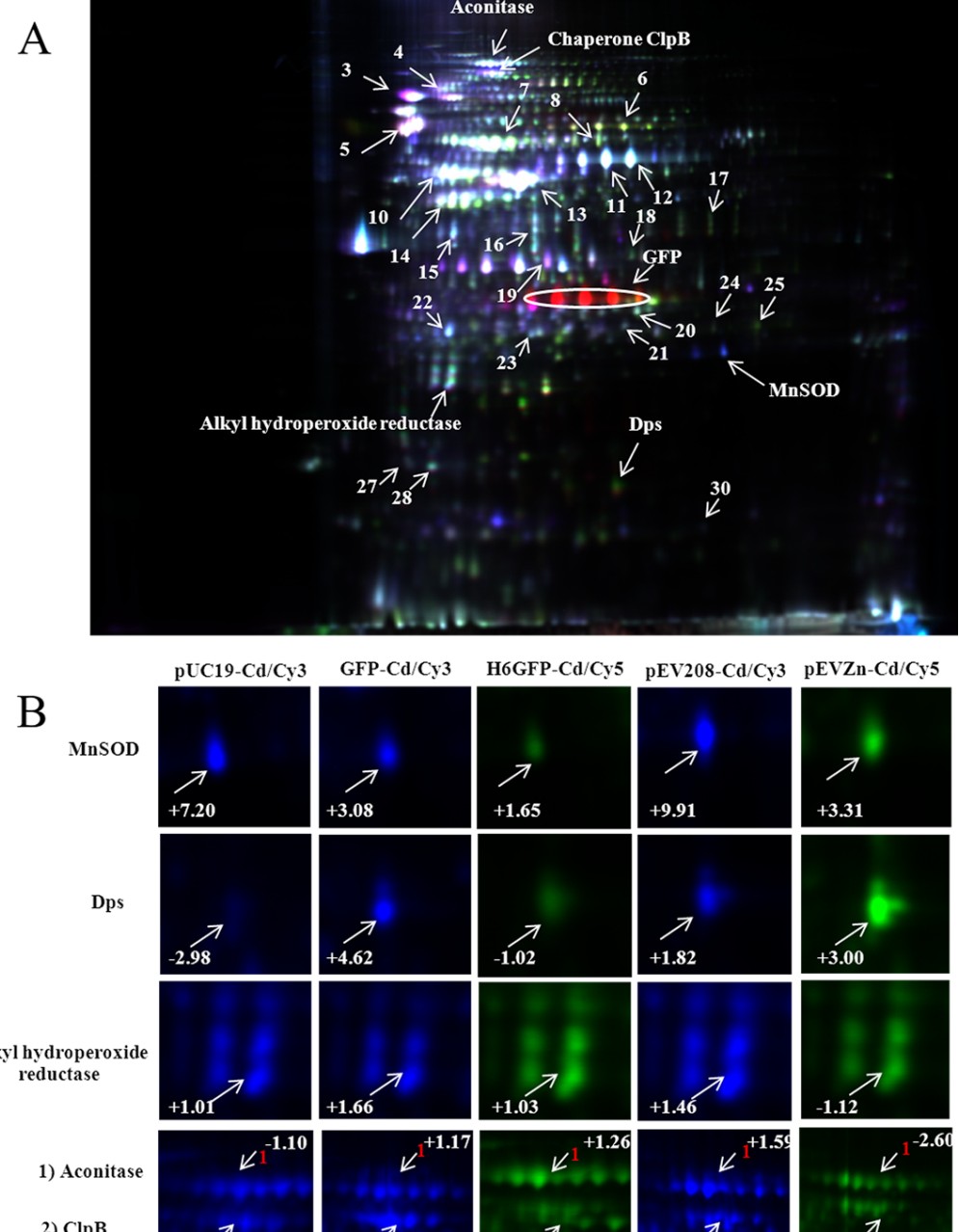

**Figure 7** **Quantitative proteomics analysis of proteins via 2D-DIGE.** Two different samples were individually labeled with two kinds of fluorescent dyes (Cy3 and Cy5) and co-separated in one gel (A). An internal standard, derived from a mixture of all samples pooled together and labeled with Cy2, was created to facilitate the normalization of each spot among all gels. Protein spots with changes in abundance ratio of 1.5 fold and $P$ values < 0.05 (Student's $t$-test) were identified and compared between the gels (B).

experimental efficiency (*Westermeier & Scheibe, 2008*). Up- or down-regulation of proteins was compared *via* the detection of fluorescence intensities of Cy3 and Cy5 simultaneously at different wavelengths. Different expression of three key proteins (MnSOD, AhpC and Dps) involved in the stress defense mechanism was selected for further investigation (shown in Fig. 7B). Intracellular complexation of cadmium ions by His6GFP rendered cells to express minute amount of MnSOD (+1.65 fold) as compared to those of TG1 carrying pUC19 (+7.20 fold) and cells expressing native GFP (+3.08 fold). A similar observation was found with regards to the compensation of MnSOD expression by the His-OmpA (+3.31 fold) as compared to cells expressing only OmpA (+9.91 fold). The same phenomenon on the expression of AhpC in cells expressing native GFP and cells expressing OmpA was observed (Fig. 7B and Table 2). The high discriminative power of DIGE could quantify the up-regulation of Dps protein of up to 4.62, 1.82 and three fold in TG1 expressing native GFP, OmpA and His-OmpA, respectively. Moreover, the quantification of aconitase expression was obtained in a tangible manner. Down-regulation of aconitase of approximately 2.60 fold was detected in the case of TG1 expressing His-OmpA upon exposure to cadmium ions. Cadmium ions also induced the expression of ClpB of approximately 2.38 fold in TG1 expressing only OmpA. On the contrary, down-regulation of ClpB of up to 2.03 fold was observed when cells expressed polyhistidine on their surface membrane.

## Interconnection between proteins implicated in cadmium stress

Analysis using the STRING software provided insights on the biological inference of proteins implicated in cadmium stress. As shown in Fig. 8, most of the important proteins as identified by 2-DE (Figs. 4, 6, 7 and Table 2) were mapped onto the protein network. MnSOD encoded by the sodA gene is known to be regulated by the SoxRS regulon (Fig. 9). Even though, alkyl hydroperoxide reductase (coded by AhpC gene) is under the control of the OxyR regulon, however, these two enzymes have some cross-functional roles and have also the same connection point as mediated by the trxA gene (coding for thioredoxin). In addition, these enzymes also function together with the F0F1 ATP synthase (an enzyme that regulates proton transportation across the membrane and ATP synthesis/hydrolysis) linked *via* the trxA gene. MnSOD was also found to share connectivity with aconitase enzyme (coded by AcnA and AcnB). Particularly, aconitases serve as protective buffers against oxidative stress. AcnA enhances the stability of the sodA transcript whereas AcnB lowers its stability (*Han & Lee, 2006*). The OxyR also plays an important role on the regulation of DNA starvation/stationary phase protection protein (Dps). The Dps unspecifically binds and protects DNA from oxidative damage mediated by hydrogen peroxide (*Han & Lee, 2006*). Moreover, ClpB, which is known as a stress-induced multi-chaperone system, was also found to possess linkages with these proteins and enzymes. Interconnections between these proteins/enzymes provide new knowledge on the cellular responses of bacteria in the presence of cadmium stress.

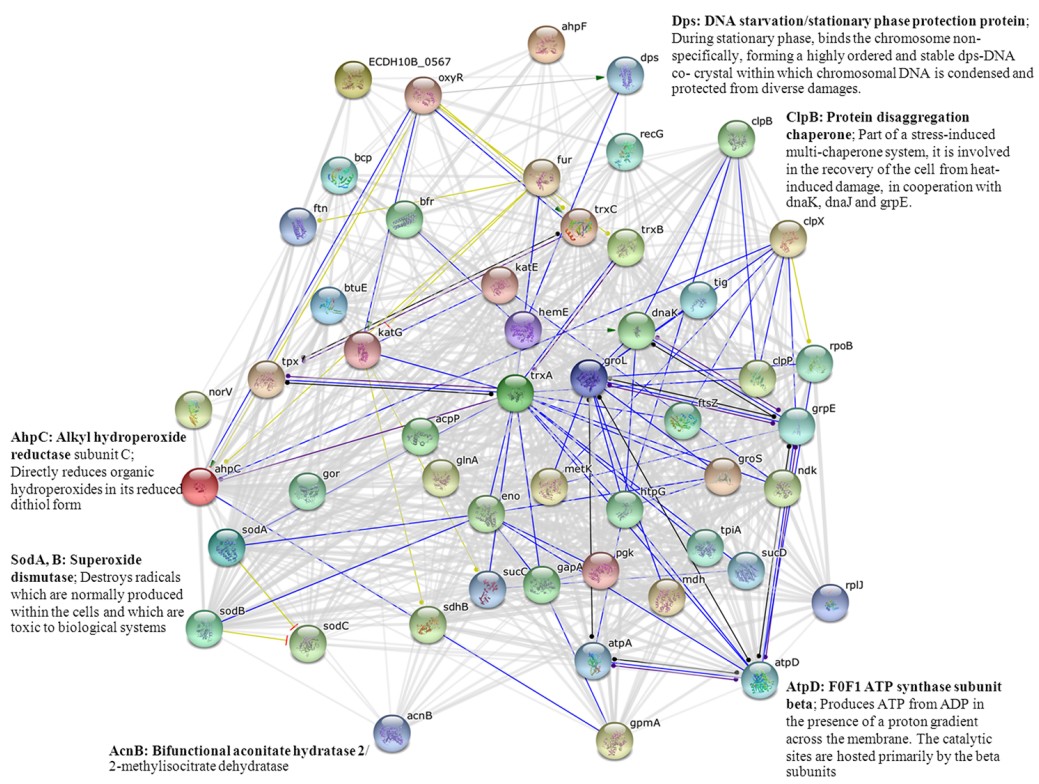

**Dps: DNA starvation/stationary phase protection protein;** During stationary phase, binds the chromosome non-specifically, forming a highly ordered and stable dps-DNA co- crystal within which chromosomal DNA is condensed and protected from diverse damages.

**ClpB: Protein disaggregation chaperone**; Part of a stress-induced multi-chaperone system, it is involved in the recovery of the cell from heat-induced damage, in cooperation with dnaK, dnaJ and grpE.

**AhpC: Alkyl hydroperoxide reductase subunit C;** Directly reduces organic hydroperoxides in its reduced dithiol form

**SodA, B: Superoxide dismutase**; Destroys radicals which are normally produced within the cells and which are toxic to biological systems

**AcnB: Bifunctional aconitate hydratase 2/ 2-methylisocitrate dehydratase**

**AtpD: F0F1 ATP synthase subunit beta**; Produces ATP from ADP in the presence of a proton gradient across the membrane. The catalytic sites are hosted primarily by the beta subunits

**Figure 8** Relationship between genes encoding proteins of *E. coli* involved in responses to cadmium stress as identified by the STRING software.

# DISCUSSION

Herein, alterations of protein expression profiles of *E. coli* as consequences of cadmium stress have successfully been investigated using 2-DE and 2D-DIGE in conjunction with mass spectrometry-based protein identification. Differentially expressed proteins of TG1 expressing metal-binding regions in the cytoplasm and on the surface membrane provided insights into the underlying mechanisms of cellular adaptation against deleterious effects of cadmium ions (Fig. 9). In normal condition, $Cd^{2+}$ is taken up by *E. coli* cells by an active transport system, in particular $Mn^{2+}$ uptake system (*Laddaga & Silver, 1985*). Inside the cells, $Cd^{2+}$ induces cell damage by generation of reactive oxygen species (ROS) through the electron transport chain (*Pacheco et al., 2008*; *Thomas & Benov, 2018*). The presence of superoxide radical ($O_2^-$) subsequently triggers the SoxRS regulon and results in the increased expression of antioxidative enzymes, mainly superoxide dismutase to protect the cells from cadmium stress (*Geslin et al., 2001*). In addition, the ROS also induces the OxyR and PerR regulons, which control the expression of catalase, peroxidases, alkyl hydroperoxide reductase enzymes. Toxic cadmium also stimulates the synthesis of cadmium-induced proteins (CDPs) such as DnaK, ClpB, RecA, UspA to form the cadmium stress stimulon (*Ferianc, Farewell & Nyström, 1998*; *Han & Lee, 2006*).

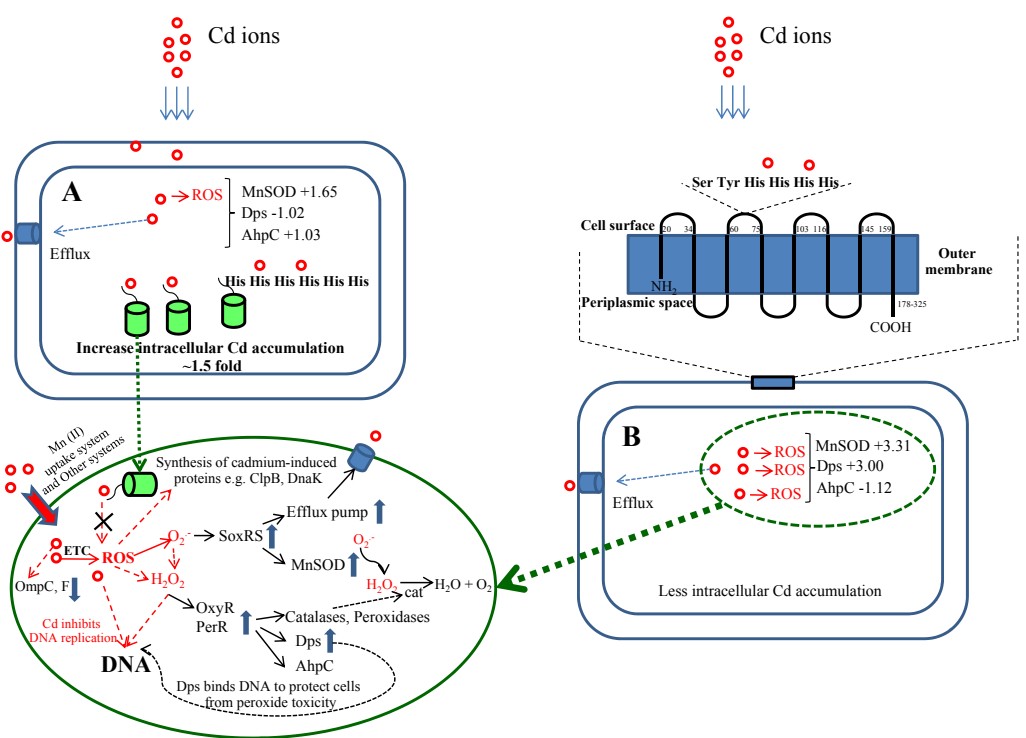

**Figure 9  A proposed molecular mechanism of cellular responses and adaptation in the presence of toxic cadmium ions.** Responses against cadmium of *E. coli* cells expressing cytoplasmic His6GFP (A) and cells expressing His-OmpA (B).

This study proposes a detailed explanation on the adaptive mechanisms of the *E. coli* host in response to toxic cadmium ions. From our findings, it can be suggested that cadmium afforded the growth arrest of control cells (TG1 harboring pUC19) (Fig. 2A). In order to survive, *E. coli* cells adapted themselves in several ways as follows:

(i)   Up-regulation of $H^+$-transporting ATPase and ATP synthase F1 for controlling proton or ions translocation as well as for ATP production.

(ii)  Reduction of OmpA form 1 together with minor changes of the OmpC precursor and OmpF were detected since these outer membrane proteins were reported to participate in the homeostasis of divalent metal ions ($Cd^{2+}$, $Zn^{2+}$ and $Cu^{2+}$) (*Egler et al., 2005*; *Faber, Egli & Harder, 1993*).

(iii) Down-regulation of oligopeptide-binding protein might also be potentiated in order to reduce the uptake of cadmium ions into cells. It is known that the oligopeptide-binding protein is a component of the oligopeptide permease and the binding protein-dependent transport system (*Han & Lee, 2006*). In *E. coli*, the oligopeptide-binding protein (coded by the OppA gene) mainly functions as a mediator for the recycling of cell wall peptides (*Wu & Mandrand-Berthelot, 1995*). The protein displays quite a broad range of substrate specificity in which it can transport any peptide of a given length with a wide variety of amino acid composition. In some circumstances, the expression of oligopeptide-binding protein in *E. coli* also increased the uptake of

aminoglycoside antibiotics (*Acosta et al., 2005*). Deletion of the OppA gene resulted in a decreased sensitivity against the aminoglycoside (*Kashiwagi et al., 1992*). Moreover, it cannot be excluded that the decreased amount of the oligopeptide-binding protein might be a result of oxidative stress. Since it has been reported to be one of the major oxidatively-damaged protein targets of *E. coli* in the presence of oxidative stress and iron overload (*Tamarit, Cabiscol & Ros, 1998*).

(iv) Once cadmium ions crossed the membranes, aconitase B was found to be markedly decreased and damaged due to its unusual FeS cluster (*Helbig, Grosse & Nies, 2008*). Cadmium ions could also trigger the SoxRS regulon and consequently leads to the increased expression of MnSOD (+7.20 fold), which is involved as the primary stress defense mechanism. Our finding was in good agreement with other studies at the transcriptional and translational levels. It has previously been reported that the adapted strain of TG1, which could tolerate cadmium ions, exhibited different transcriptional profiles from the sensitive strain (*Brocklehurst & Morby, 2000*). The intracellular generation of superoxide radical by cadmium (*Thomas & Benov, 2018*), nickel and cobalt exerted toxic effects toward *E. coli* in which the participation of *sod* genes was regulated for the protection against metal stress (*Geslin et al., 2001*). Increased production of the SOD enzyme was found in the *E. coli* strain BL21 and P4 in the presence of cadmium (*Khan et al., 2017*; *Shen et al., 2012*).

(v) In parallel, proteins involved in energy metabolism, e.g., glyceraldehyde-3-phosphate dehydrogenase and glycerol kinase, were down-regulated in order to reduce the energy consumption as required for cell survival. Therefore, the reduction of the oligopeptide-binding protein (mentioned in iii) and amino acid ABC transporter substrate-binding protein was observed for regulating the transportation of amino acid and other carbon sources during energy crisis (*Isarankura-Na-Ayudhya et al., 2009*).

To further confirm the aforementioned mechanisms, GFP harboring hexahistidine (His6GFP) was included to mediate the intracellular cadmium complexation (Fig. 9). Interestingly, low level expression of MnSOD (+1.65 fold) and AhpC (+1.03 fold) was detected. On the contrary, the up-regulation of both antioxidant enzymes (+3.08- and +1.66 fold) was found in cells expressing only native GFP (Fig. 7B). Our findings indicated that the effective complexation of cadmium ions by histidine residue provided protective roles against toxic metals, possibly by inhibition of ROS production. It has been detected that the presence of His6GFP rendered the cells to accumulate cadmium ions intracellularly up to $8.00 \pm 0.08$ nmole/$8 \times 10^8$ cells or ~1.5 fold higher than those of cells expressing native GFP ($5.35 \pm 0.45$ nmole/$8 \times 10^8$ cells) and TG1 host ($5.34 \pm 0.7$ nmole/$8 \times 10^8$ cells). Such protection coincided with the recovery of growth characteristics of His6GFP-expressing cells in the presence of cadmium (Fig. 2C). However, it should be noted that the overexpression of native GFP somewhat protected the *E. coli* cells from toxic cadmium (Fig. 2B). Plausible explanations might be attributable to (i) the quenching ability and SOD-like activity of the native GFP to detoxify the superoxide radical (*Bou-Abdallah, Chasteen & Lesser, 2006*) and (ii) the increased transcription (*Helbig, Grosse & Nies, 2008*) and translation of alkyl hydroperoxide reductase (AhpC) (Table 2 and Fig. 7B) under the activation of OxyR. This enzyme belongs to a large family of thiol-specific antioxidant

proteins found in both prokaryotes and eukaryotes. It is known to detoxify not only peroxides but also reactive oxygen-, nitrogen- and sulfur-species. Cloning of this enzyme from *Anabaena* spp. in *E. coli* rendered cells to be more resistant against $CdCl_2$ of up to 4 mM (*Mishra, Chaurasia & Rai, 2009*). In other organisms, increased levels of alkyl hydroperoxide reductase protected the plant pathogenic bacterium namely *Xanthomonas campestris* from peroxides and cadmium stress (*Banjerdkij, Vattanaviboon & Mongkolsuk, 2005*). Mutant cells of *Helicobacter pylori* defective in AhpC were more sensitive to oxidative stress conditions (*Wang et al., 2005*). In *Saccharomyces cerevisiae*, disruption of the AHP1 gene rendered cells to be more susceptible to several kinds of metal ions (*Nguyen-nhu & Knoops, 2002*).

Next, the blockage of metal uptake as a consequence of anchored polyhistidine residues on surface membrane has also been proven to confer protection against cadmium toxicity in *E. coli* (Fig. 5). Such protective effect is comparable to that observed in cells expressing His6GFP (Fig. 2C). Results from proteomics analysis revealed differences of protein expression profiles among these two cases as follows. Up-regulation of MnSOD, glyceraldehyde-3-phosphate dehydrogenase form 2, ribosomal protein L9 and OmpC precursor was found in cells expressing His-OmpA. An important item of note is that down-regulation and no increased expression of ATP synthase F1 and $H^+$-transporting ATPase were observed. This infers that the binding of cadmium ions was effectively potentiated at the surface membrane in which the regulation of ions-transporting system was not induced. However, a minute amount of cadmium ions could pass through the membrane portion and exert their toxic effects to cells as observed by the increased expression of MnSOD. Moreover, expression of the OmpC precursor (membrane protein porin) proportionally corresponded with information from transcriptional profiles of Cd-resistant strain of *E. coli* (*Brocklehurst & Morby, 2000*). Meanwhile, a mutant strain lacking the OmpC gene was proven to be sensitive to cadmium ions (*Egler et al., 2005*). In another case, the expression of OmpA alone could also protect *E. coli* cells from cadmium-induced growth arrest (Fig. 5). However, the induction of antioxidative scavenging enzyme consisting of AhpC and ATP synthase F1 was still required (Table 2 and Fig. 7B). These results were in contrast to those observed in cells expressing His6GFP (Table 2 and Fig. 7B). Therefore, all findings lend support to the notion that metal complexation by cytoplasmic metal-binding protein afforded more efficiency to cope with cadmium stress than the metal-binding affinity at the surface membrane (Fig. 9).

## CONCLUSION

This study explores the underlying mechanism and cellular responses of bacteria against toxic cadmium ions. In particular, *E. coli* TG1 expressing His6GFP and His-OmpA were used as models for studying the roles of cytoplasmic metal complexation and metal chelation at the surface membrane, respectively, upon exposure to cadmium stress (Fig. 9). Results from 2-DE and 2D-DIGE together with mass spectrometry-based protein identification had revealed that the complexation of cadmium ions by His6GFP helped to reduce toxicity of cadmium-induced oxidative cell damage *via* initial inhibition of

ROS production. Supportive evidences could be accounted mainly by the low expression level of antioxidative enzymes and stress responsive proteins under the regulations of SoxRS and OxyR as compared to those derived from the effect of His-OmpA. Thus, it can be rationalized that the complex formation between cadmium ions and histidine-rich peptides/proteins provides more efficiency to cope with cadmium stress than the blockage of metal uptake at the surface membrane. Such findings shed light on the molecular mechanism and cellular adaptation of bacteria against cadmium toxicity.

### Funding

This research is financially supported in part by the Young Research Scholar grant (no. MRG 5480205) from the Thailand Research Fund to Patcharee Isarankura-Na-Ayudhya and the annual governmental grant of Mahidol University under the National Research Universities Initiative. There was no additional external funding received for this study. The funders had no role in study design, data collection and analysis, decision to publish, or preparation of the manuscript.

### Grant Disclosures

The following grant information was disclosed by the authors:
Young Research Scholar: MRG 5480205 Thailand Research Fund.
Mahidol University under the National Research Universities Initiative.

### Competing Interests

The authors declare there are no competing interests.

### Author Contributions

- Patcharee Isarankura-Na-Ayudhya conceived and designed the experiments, performed the experiments, analyzed the data, contributed reagents/materials/analysis tools, prepared figures and/or tables, authored or reviewed drafts of the paper, approved the final draft.
- Chadinee Thippakorn, Nipawan Bunmee and Suchitra Sawangnual performed the experiments, authored or reviewed drafts of the paper, approved the final draft.
- Supitcha Pannengpetch performed the experiments, analyzed the data, prepared figures and/or tables, authored or reviewed drafts of the paper, approved the final draft.
- Sittiruk Roytrakul performed the experiments, analyzed the data, contributed reagents/materials/analysis tools, authored or reviewed drafts of the paper, approved the final draft.
- Chartchalerm Isarankura-Na-Ayudhya conceived and designed the experiments, analyzed the data, prepared figures and/or tables, authored or reviewed drafts of the paper, approved the final draft.
- Virapong Prachayasittikul analyzed the data, contributed reagents/materials/analysis tools, authored or reviewed drafts of the paper, approved the final draft.

## Data Availability

The raw data are provided in the Supplementary File.

## Supplemental Information

Supplemental information for this article can be found online at http://dx.doi.org/10.7717/peerj.5245#supplemental-information.

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
