# Peer review of "Metal complexation by histidine-rich peptides confers protective roles against cadmium stress in Escherichia coli as revealed by proteomics analysis"

_PeerJ, doi:10.7717/peerj.5245_

## Round 0.1 · original submission · Major Revisions

· Academic Editor

Major Revisions

Dear Chartchalerm

The Reviewer's evaluations are now available. Please respond to all their points in a revision.

·

Basic reporting

The authors of this manuscript have clearly stated the biological significance of this study. The authors proposed a sound hypothesis of the molecular mechanism that governs the adaption of E. coli to toxic cadmium stress. The manuscript provided sufficient background information with appropriate literature references. The whole information provided in this manuscript was well structured; the figures and tables were properly displayed for supporting.

This reviewer suggests adding another figure to better describe the authors' conclusion. The authors described a network of protein that are regulated under the stress of cadmium and the expression of PolyHistidine. A picture of this network would be a nice summary for the proposed mechanism of cadmium adaptation.

Experimental design

This manuscript represents an article for an original primary research within the aims and scope of PeerJ. Understanding the mechanism for bacterial in response to toxic ion strength is of significance in Microbiology, Ecology and other scientific fields. E. coli, used in this report represents an excellent target at lease because it is a convenient and relevant organism to deal with in a laboratory setting. The authors have conducted proper experimental design. The experimental data are sufficient and unambiguous to support the authors' hypothesis and to facilitate further studies in this topic.

Validity of the findings

no comment

·

Basic reporting

Work is a labor intensive effort. The presentation and content are nicely lined up.

Experimental design

Experimental design is presented very well.

Validity of the findings

The results of the work are well presented. However, since the results of E. coli in different expression properties are close to each other, what is the effect of the measurement uncertainty of the study on these results? It is unknown. This should be well explained.

***I strongly recommend evaluating these results by a scientist working in proteomic field.

Additional comments

Work is a labor intensive effort. The presentation and content are nicely lined up.
The results of the work are well presented. However, since the results of E. coli in different expression properties are close to each other, what is the effect of the measurement uncertainty of the study on these results? It is unknown. This should be well explained.

Reviewer 3 ·

Basic reporting

This manuscript describes the use of cytoplasmic and membrane bound polyhistidine derivatives for bacterial adaptation to toxic cadmium ions. Using susceptibility assays and proteomics, this study compares effects on bacterial growth and protein synthesis.

The manuscript has a lot of scope for improvement with regard to grammar, sentence structure and scientific accuracy. Some examples of areas to improve are listed below:

Abstract

Lines 39-42: the main point is not clear. Please rewrite these lines to better describe the results related to expression of key proteins.
Line 43: “minute expression of MnSOD…”, relative to what?

Introduction

Line 55 – Opening sentence is missing references.
Line 56 – More specific descriptors of the effects of Cadmium ions. For e.g., “breakage” of DNA, state the nature of alteration of cell division (slower, faster, dysregulated, etc).
Line 63 – The knowledge gap is not clear. The authors describe several mechanisms of metal toleration, e.g. reduction of uptake, sequestration, efflux pumps, yet state that “the underlying mechanism of metal toleration in bacteria remains unclear…”/
Line 69 – The meaning of “accumulation” is not clear.
Lines 71-73 – please revise so as to improve sentence structure.
Line 87 – describe the specific mechanism being referred to. Do all these proteins adopt the same mechanism?
Line 94 – replace ‘prove’ with a more appropriate term, such as ‘test’ or ‘determine’.

The Introduction is missing a description of the current understanding of the toxic effects of cadmium ions on E coli, mechanisms of cellular response and adaptation. I urge the authors to conduct a thorough literature survey. (A simple Pubmed search for the keywords ‘cadmium toxicity bacteria’ led to more than 900 hits.) Moreover, the authors should provide a strong rationale for studying protein expression in their model strains. How does cadmium ion treatment affect protein expression profiles?

Materials and Methods

Line 110 – host ‘cells’ should be singular.
Line 126 – Comment on the choice of Cadmium dose. Did the authors perform a dose response assay?

Experimental design

Results

Line 221 – Title includes ‘growth pattern’ but the following text does not discuss it.
Line 229 – For this experiment and Figure 2, report the doubling time, i.e. time required by cells to double their optical density, for all strains and conditions. Presenting this analysis in a table format might be more informative to readers.
Line 233 – Please explain the choice of Cadmium ion dose (0.2 mM). Did the authors perform a kill assay to determine susceptibility/survival of all relevant E coli strains used in this work, as a function of Cadmium dose? This is essential to rationalize the choice of dose for all the experiments described in this manuscript.
Line 235-236 – The assertion that the presence of hexahistidine-GFP promotes cell growth is merely correlative at this point. Please rewrite this sentence to better describe the observation. For example, ‘Cells expressing the hexahistidine-GFP fusion protein displayed similar growth rates in the presence and absence of Cadmium ions.’
Lines 246-263 – These lines describe changes in protein expression profile in the three strains as a function of cadmium ion stress. I urge the authors to report on the total number of proteins identified in each strain and condition, including common proteins identified in more than 1 strain and condition. For each protein, state the fold-change values while comparing expression profile changes in the various strains. Some of this information is provided in Table 2, but including the numbers in the results section will help readers.
Line 256 – Please be specific in reporting changes observed in expression profiles. For example, ‘H+ transporting ATPase was found to be expressed in an opposite manner.’ Explain what the ‘opposite manner’ is.
Line 263 – ‘… while OmpF was found in all cases in response to cadmium ions.’ The point of this line is not clear. Do the authors mean that OmpF was identified in all cases? Or was the trend of fold-change the same in all cases?
Lines 273-274 and Figure 5 – Please state the number of replicate experiments performed to generate this data. Figure 5 is lacking error bars. Please represent OD600 data as mean +/- standard error of at least 3 biological replicates.
Overexpression of OmpA results in similar growth rates in the presence and absence of cadmium ions. The increase in growth levels observed in the case of polyhistidine-OmpA is small. It remains to be seen whether the difference in growth rate/levels between OmpA and hexahistidine-OmpA are within the error of the experiment. I urge the authors to be cautious while interpreting any differences in the growth rate of the two strains. Based on the data, the contribution of the hexahistidine tag to Cadmium resistance is unclear to me. Also, it is important to provide proper context to these results relative to a wild-type control carrying the empty vector.
Line 284 – needs appropriate referencing.
Line 290 – ‘was disappeared’ may be replaced with a more appropriate term, such as ‘was missing/not detected’.
Line 303 – ‘expression’ should be replaced with ‘overexpressing’
Lines 288-312 – for the fold-change trends reported in this section, I urge the authors to explain their use of descriptors such as ‘a remarkable’ expression… (line 301), MnSOD became ‘the most important’ target… (line 303). Discuss fold changes observed in cells expressing native GFP – what is the explanation for differences in expression between cells expressing native GFP and cells carrying the empty vector (for example, MnSOD expression, line 330)?
Line 387 – OppA should be OmpA?

Line 318 – ‘great discrimination’ – consider alternative terminology

Line 432-434 – I disagree with the authors on this conclusion. Figure 5 shows that cells overexpressing the polyhistidine OmpA exhibit slightly elevated growth rate/level relative to cells overexpressing OmpA. As mentioned in a previous comment, this growth curve should be replotted to include average data from at least three independent biological replicates (mean +/- standard error). Moreover, a growth curve alone provides little information on the mechanism of cytoprotection, if any. It is preliminary and inaccurate to suggest that this experiment proves that inhibition of metal uptake by the polyhistidine residues confers resistance to cadmium toxicity.

To further investigate the mechanism of cytoprotection mediated by the polyhistidine moiety, I suggest the authors measure intracellular levels of Cadmium ions in all the strains, under all the treatments, described in this work. Similar studies have been performed by Sousa et al (Nature Biotechnology 1996), who used hexa-histidine conjugated LamB, an outer membrane protein in E coli. The authors should explain their choice of OmpA for tagging with hexahistidine, as opposed to LamB (as in Sousa et al).

2D-DIGE – Was reciprocal labeling of the samples performed, to eliminate artifactual spots due to dye-dependent interactions?

Validity of the findings

Lack of relevant controls

The susceptibility and proteomics experiments are missing appropriate positive controls, i.e. strains that have been demonstrated previously to affect Cadmium ion toxicity (for e.g. hexa-histidine LamB in Sousa et al). Use of a positive control is essential to provide context to the effect of unknown alterations, such as poly histidine GFP and OmpA. Moreover, the OmpA experiments are missing a control strain that does not overexpress OmpA. This must be included in any repeat experiments because OmpA itself facilitates transport of ions and solutes, therefore, its overexpression might alter transport of cadmium ions across the membrane. I suggest the authors repeat their experiments to incorporate relevant positive controls.

The fold-change in expression of OmpA in the strains described in the text should be measure relative to native expression in the appropriate parent strain. Plasmid based overexpression is often accompanied by misfolded or aggregated proteins, such as inclusion bodies. The correlation between expression and activity of OmpA needs to be validated.

It will be important to compare the susceptibility and proteomics results obtained in hexahistidine-expressing strains with previous regulators of cadmium toxicity such as OxyR, SoxRS, RpoH, etc, which are already outlines in the Introduction (Line 86). The proteomic signature, in particular, might reveal important mechanisms of response based on these previously known models.

Additional comments

This study provides new information on the effects of cadmium ion treatment in bacteria. The findings are undoubtedly relevant to several areas of microbiology and biotechnology. I urge the authors to incorporate control strains to provide proper context to their findings. Specific suggestions in this regard are provided in my comments. Additionally, significant improvements can be made to all sections of the manuscript, this will surely elevate the impact of this work.

---

## Round 0.2 · accepted · Accept

· Academic Editor

Accept

Dear Chartchalerm,

Thank you for your submission to PeerJ. I am writing to inform you that your manuscript - Metal complexation by histidine-rich peptides confers protective roles against cadmium stress in Escherichia coli as revealed by proteomics analysis - has been Accepted for publication.

Best regards
Nuri Azbar

# ·

Basic reporting

Work is a labor intensive effort. The presentation and content are nicely lined up.

Experimental design

Experimental design is presented very well.

Validity of the findings

Explanations about the experimental design of the article by the researchers regarding the criticisms are sufficient.

Additional comments

I recommend the paper for publication. In my opinion, the paper should be published in PeerJ.